# Rethinking Impersonation and Dodging Attacks on Face Recognition Systems

## ABSTRACT

Face Recognition (FR) systems can be easily deceived by adversarial examples that manipulate benign face images through imperceptible perturbations. Adversarial attacks on FR encompass two types: impersonation (targeted) attacks and dodging (untargeted) attacks. Previous methods often achieve a successful impersonation attack on FR; However, it does not necessarily guarantee a successful dodging attack on FR in the black-box setting. In this paper, our key insight is that the generation of adversarial examples should perform both impersonation and dodging attacks simultaneously. To this end, we propose a novel attack method termed as Adversarial Pruning (Adv-Pruning), to fine-tune existing adversarial examples to enhance their dodging capabilities while preserving their impersonation capabilities. Adv-Pruning consists of Priming, Pruning, and Restoration stages. Concretely, we propose Adversarial Priority Quantification to measure the region-wise priority of original adversarial perturbations, identifying and releasing those with minimal impact on absolute model output variances. Then, Biased Gradient Adaptation is presented to adapt the adversarial examples to traverse the decision boundaries of both the attacker and victim by adding perturbations favoring dodging attacks on the vacated regions, preserving the prioritized features of the original perturbations while boosting dodging performance. As a result, we can maintain the impersonation capabilities of original adversarial examples while effectively enhancing dodging capabilities. Comprehensive experiments demonstrate the superiority of our method compared with state-of-the-art adversarial attacks.

## CCS CONCEPTS

• **Computing methodologies → Biometrics**.

## KEYWORDS

Face Recognition, Adversarial Attacks, Adversarial Attacks on Face Recognition, Impersonation Attacks, Dodging Attacks

## 1 INTRODUCTION

Thanks to the ceaseless advancements in deep learning, Face Recognition (FR) has achieved exceptional performance [1, 2, 9, 26, 41, 49]. However, the vulnerability of existing FR models to adversarial attacks poses a significant threat to their security. Hence, there is an urgent need to enhance the performance of adversarial face

Permission to make digital or hard copies of all or part of this work for personal or classroom use is granted without fee provided that copies are not made or distributed for profit or commercial advantage and that copies bear this notice and the full citation on the first page. Copyrights for components of this work owned by others than the author(s) must be honored. Abstracting with credit is permitted. To copy otherwise, or republish, to post on servers or to redistribute to lists, requires prior specific permission and/or a fee. Request permissions from permissions@acm.org.

*ACM MM, 2024, Melbourne, Australia*

© 2024 Copyright held by the owner/author(s). Publication rights licensed to ACM.
ACM ISBN 978-x-xxxx-xxxx-x/YY/MM
https://doi.org/10.1145/nnnnnnn.nnnnnnn

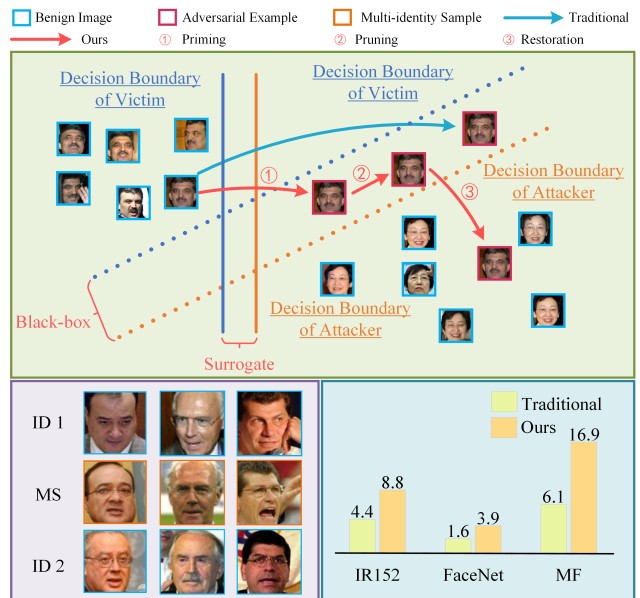

**Figure 1: Top: previous methods that achieve a successful impersonation attack on FR cannot guarantee a successful dodging attack on FR in the black-box setting. In contrast, we present Adv-Pruning, including Priming, Pruning, and Restoration Stages, to perform both impersonation and dodging attacks simultaneously. Bottom (left): natural Multi-identity Samples (MS). Bottom (right): the dodging Attack Success Rate (%) between the previous methods and Adv-Pruning on multiple models.**

examples to expose more blind spots in FR models. As a result, several research endeavors have been directed towards this realm. A multitude of adversarial attacks have been developed to create adversarial face examples with characteristics such as stealthiness [7, 18, 40, 42, 61], transferability [29, 69–71], and physical attack capability [28, 62, 63]. These efforts contribute to enhancing the effectiveness of adversarial attacks on FR. Nevertheless, these studies primarily concentrate on bolstering either impersonation attacks or dodging attacks, overlooking the exploration of the effectiveness of dodging attacks when crafting adversarial face examples using impersonation attacks.

In real-world deployment contexts, individuals with malicious intent are prone to creating adversarial face examples incorporating their own facial features to manipulate FR systems to mistakenly identify them as pre-defined victims during impersonation attacks. Concurrently, the individuals strive to evade accurate identification as perpetrators, thereby circumventing detection and preventing legal accountability. This requires the creation of adversarial examples capable of executing both impersonation and dodging attacks

simultaneously. In the realm of adversarial attacks on image classification, a successful impersonation attack typically implies a successful dodging attack. However, FR is an open-set task [9, 49], which is quite different from image classification. In the real-world deployment of FR systems, accurately predicting the class probability of identities presents an extreme challenge. Therefore, we extract embeddings from two face images using the FR model. Subsequently, the distance between the two embeddings is used to determine whether the images belong to the same identity. If the distance falls below a predefined threshold, the two images are recognized as belonging to the same identity; otherwise, they are classified as different identities. Based on the measurement of FR, there are two decision boundaries for each FR model when crafting adversarial examples as shown in Fig. 1. As a result, there exists the natural samples that can be classified as two different identities in theory. We denote these samples as *multi-identity samples* (refer to Section 4.4).

The existence of multi-identity samples implies that a successful impersonation attack on FR does not necessarily guarantee a successful dodging attack on FR. Existing research indicates that an adversarial sample was located near the decision boundary [4, 17]. Suppose we generate adversarial face examples using previous methods. In the white-box setting, both the structures and parameters of the victim models are known, enabling the generation of adversarial face examples that can cross the decision boundaries of both attacker and victim, as shown in Fig. 1. However, in the black-box setting, the decision boundaries of black-box models differ from those of the surrogate models. Consequently, adversarial examples generated on the surrogate model lie near the decision boundary of the victim, preventing them from crossing the decision boundary of the attacker. As such, the majority of adversarial face examples crafted by previous methods, which can successfully perform impersonation attacks, fail to perform dodging attacks in the black-box setting.

In this paper, we propose a novel attack method, termed as Adversarial Pruning (Adv-Pruning). In the realm of adversarial attacks on FR, previous impersonation methods have achieved a significant level of sophistication. However, there remains a pressing need to bolster the efficacy of adversarial face examples in dodging attacks. Consequently, our research is directed towards enhancing the dodging attack performance of adversarial face examples while maintaining the impersonation attack performance. Specifically, we introduce an attack consisting of three stages: Priming, Pruning, and Restoration. In the Priming stage, we optimize the adversarial examples to ensure adequate attack potential. In the Pruning stage, with considering the pruning concept in model compression, we propose Adversarial Priority Quantification to measure the region-wise priority of original adversarial perturbations using an priority measure which is directly proportional to the supremum of the absolute model output variances. After processing by Adversarial Priority Quantification, we prune the adversarial face examples to free up less prioritized adversarial perturbations. In the Restoration stage, we propose Biased Gradient Adaptation to add biased gradient perturbations favoring dodging attacks on the pruned regions to adapt the adversarial face examples into the space that can be classified as the victim while remaining unidentifiable as the attacker, thereby enhancing the dodging performance of the

adversarial face examples without compromising the prioritized features of original adversarial perturbations. As illustrated in the top of Fig. 1, after undergoing these stages, the adversarial face example generated by our proposed method can successfully traverse the decision boundaries of both the attacker and victim of the black-box model, achieving successful black-box impersonation and dodging attacks.

Our main contributions are summarized as follows:

- We offer a new perspective for adversarial attacks on FR models that the generation of adversarial examples should perform both impersonation and dodging attacks simultaneously. To the best of our knowledge, this is the first work that studies the universality of multi-identity samples among adversarial face examples crafted by impersonation attacks.
- We propose a novel adversarial attack method called Adversarial Pruning (Adv-Pruning). Adversarial Priority Quantification is presented to quantify the priority of the adversarial perturbations with minimal impact on absolute model output variances. Biased Gradient Adaptation is designed to adapt the adversarial examples to traverse both the decision boundaries of attacker and victim using biased gradients.
- Extensive experiments demonstrate that our proposed method achieves superior performance compared to the state-of-the-art adversarial attack methods. Moreover, our presented method could be plugged into various FR systems and adversarial attack methods.

## 2 RELATED WORK

### 2.1 Adversarial Attacks

The primary objective of adversarial attacks is to introduce imperceptible perturbations to benign images to deceive machine learning systems and cause them to make mistakes [14, 47]. The existence of adversarial examples poses a significant threat to the security of current machine learning systems. Lots of efforts have been dedicated to researching adversarial attacks in order to enhance the robustness of these systems [12, 30, 33, 34, 37, 43, 54, 67, 68, 72]. To improve the performance of black-box adversarial attacks, DI [59] applies random transformations to adversarial examples in each iteration to achieve a data augmentation effect. VMI-FGSM [51] employs gradient variance to stabilize the updating process of adversarial examples, boosting the black-box performance. SSA [33] transforms adversarial examples into the frequency domain and uses spectrum transformation to augment them. SIA [53] applies a random image transformation to each image block, generating a varied collection of images that are then employed for gradient calculation. BSR [50] divides the input image into multiple blocks, subsequently shuffling and rotating these blocks in a random manner, creating a collection of new images for the purpose of gradient calculation. DA [15] utilizes dispersion amplification to enhance the multi-task attack capability of adversarial attacks. Despite their gratifying progress, these studies neglect the consideration of pruning adversarial examples through introducing pruning methods into the realm of adversarial attacks. In our research, we propose a novel pruning method capable of identifying and freeing up the adversarial perturbations with minimal impact on absolute model

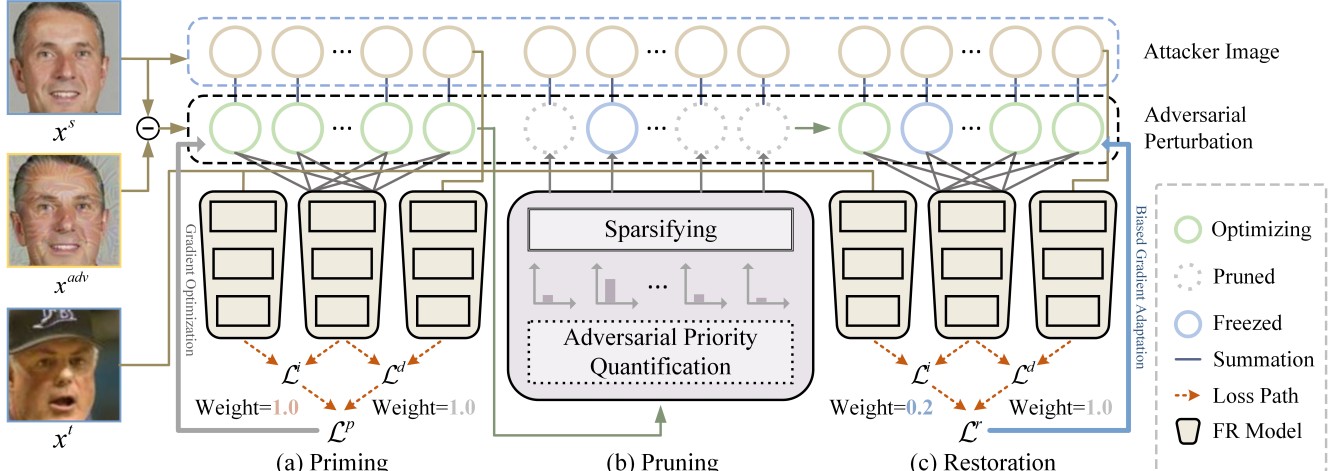

**Figure 2: Overview of our Adv-Pruning attack framework, which consists of Priming, Pruning, and Restoration stages. (a) During the Priming stage, we optimize the adversarial examples to ensure they have sufficient attack performance. (b) In the Pruning stage, we propose Adversarial Priority Quantification to quantify the priority of adversarial perturbations. Subsequently, we sparsify the adversarial perturbations based on the quantified priorities. (c) In the Restoration stage, we present Biased Gradient Adaptation to introduces gradient perturbations biased to dodging attacks on the sparsified regions.**

output variances, thereby sparsifying regions for adding adversarial perturbations with the aim of dodging capabilities improvement.

## 2.2 Adversarial Attacks on Face Recognition

Based on the restriction of the adversarial perturbations, adversarial attacks on FR can be classified into two categories: restricted attacks [5, 10, 31, 32, 35, 60, 73] and unrestricted attacks [3, 6, 8, 44, 46, 48, 55, 56, 64]. Restricted attacks on FR are the attacks that generate adversarial examples in a restricted bound (e.g. $L_p$ bound). To enhance the transferability of adversarial attacks on FR, [69] propose DFANet, which applies dropout on the feature maps of the convolutional layers to achieve ensemble-like effects. In addition, [71] introduces BPFA, which further improves the transferability of adversarial attacks on FR by incorporating beneficial perturbations [57] on the feature maps of the FR models, resulting in hard model augmentation effects. [29] leverages extra information from FR-correlated tasks and uses a multi-task optimization framework to enhance the transferability of crafted adversarial examples. The unrestricted adversarial attacks on FR are the attacks that generate adversarial examples without the restriction of a predefined perturbation bound. They mainly focus on physical attacks [28, 58, 62], attribute editing [22, 40] and generating adversarial examples based on makeup transfer [18, 42, 63]. The existing literature on both restricted and unrestricted adversarial attacks on FR systems has successfully enhanced the performance of these attacks. Nevertheless, it remains under-explored in the correlation between impersonation and dodging attacks. This paper elegantly addresses this by investigating the correlation between impersonation and dodging attacks and introducing a novel attack method that bolsters the dodging capabilities while preserving the impersonation capabilities of previous methods.

## 3 METHODOLOGY

### 3.1 Problem Formulation

Let $\mathcal{F}^{vct}(x)$ denote the FR model used by the victim to extract the embedding from a face image $x$. We refer to $x^s$ and $x^t$ as the attacker and victim images, respectively. The objective of the impersonation attacks explored in our research is to manipulate $\mathcal{F}^{vct}$ in order to misclassify $x^{adv}$ as $x^t$, while ensuring that $x^{adv}$ bears a close visual resemblance to $x^s$. By contrast, the objective of the dodging attacks proposed in this study is to render $\mathcal{F}^{vct}(x)$ unable to identify $x^{adv}$ as $x^s$, while simultaneously ensuring that $x^{adv}$ bears a visual resemblance to $x^s$. For the sake of clarity and conciseness, the detailed optimization objectives for both impersonation and dodging attacks are provided *in the supplementary*.

Few works explore the correlation between impersonation and dodging attacks on FR. In the following, we delve into the correlation between these two types of attacks and propose a novel method to enhance dodging attacks while maintaining impersonation attacks. An overview of the proposed method is illustrated in Fig. 2. As depicted in Fig. 2, our proposed method is structured into three stages: Priming, Pruning, and Restoration. Through the sequential application of these stages, we are able to generate adversarial examples that exhibit a potent combination of impersonation and dodging attack capabilities.

### 3.2 Exploring the Impersonation and Dodging Attack on Face Recognition

In most cases, the victim model $\mathcal{F}^{vct}$ is not accessible to the attacker, making it extremely challenging to optimize the objectives for black-box attacks directly. To circumvent this issue, a common approach is to leverage a surrogate model $\mathcal{F}$ accessible to the attacker to

generate adversarial examples that can be transferred to the victim model for an effective attack [5, 11, 13, 27, 38, 39, 50, 53, 65, 67].

For impersonation attacks, the loss can be formulated as follows:

$$\mathcal{L}^i = \|\phi\left(\mathcal{F}\left(x^{adv}\right)\right) - \phi\left(\mathcal{F}\left(x^t\right)\right)\|_2^2 \tag{1}$$

where $\phi(x)$ represents the operation that normalizes $x$. $x^{adv}$ is the adversarial example which is initialized with the same value as $x^s$. The loss function of dodging attacks can be formulated as:

$$\mathcal{L}^d = -\|\phi\left(\mathcal{F}\left(x^{adv}\right)\right) - \phi\left(\mathcal{F}\left(x^s\right)\right)\|_2^2 \tag{2}$$

As the FR task is an open-set task, it is impractical to predict the classes of users during the practical deployment of the FR model. Therefore, we need to compare the distance between two face images to discern whether they depict the same identity or not. Based on the identification method in FR, multi-identity samples exist theoretically. Our experiments verify the existence of such samples among benign face images. The existence of multi-identity samples raises a question:

**Does the success of an impersonation attack imply the success of dodging attacks on FR systems?**

To this end, we generate adversarial face examples using the previous impersonation attack and evaluate its dodging Attack Success Rate (ASR). Our experiment confirms that the majority of adversarial examples crafted through previous methods, which are successful in performing impersonation attacks, fail to successfully execute dodging attacks in the black-box setting (see Section 4.4).

Nonetheless, in real-world adversarial attacks, attackers do not want the adversarial face examples to be recognized as themselves, as this may lead to legal consequences. Hence, it is crucial to research attack techniques that can execute both impersonation and dodging attacks simultaneously. Previous methods on FR systems have shown a remarkably high level of impersonation ASR in black-box settings. Therefore, our objective is to enhance the dodging performance while maintaining the impersonation effectiveness of previous attack methods.

To accomplish this objective, a straightforward approach is to generate adversarial face examples using a multi-task attack strategy. In the following, we will take the Lagrangian attack strategy as the example for its simplicity. The Lagrangian attack strategy utilizes the following loss function to craft adversarial examples:

$$\mathcal{L} = \lambda \mathcal{L}^i + \mathcal{L}^d \tag{3}$$

However, due to the conflict between the optimization between $\mathcal{L}^i$ and $\mathcal{L}^d$, there exists a trade-off between the performance of impersonation and dodging performance, leading to subpar performance (See Section 4.4). Suppose we can mitigate the trade-off, we will achieve a better dodging performance while maintaining the impersonation performance.

### 3.3 Adversarial Pruning Attack

To accomplish this objective, a straightforward approach is to fine-tune the adversarial face examples generated by the Lagrangian attack with a lower $\lambda$ value in order to enhance the performance of dodging attacks. However, this method does not enhance the dodging attack performance without compromising the impersonation attack performance (see *Fine-tuning* in Table 3). We contend that

this issue arises because the newly introduced adversarial perturbation that favors dodging attacks ends up disrupting the prioritized features of existing adversarial perturbation. While it may improve the performance of dodging attacks, it inevitably diminishes the performance of impersonation attacks. To address this, we introduce new perturbations favoring dodging attacks in regions where original perturbations are not added. Nevertheless, identifying suitable areas for these new perturbations is challenging due to their scarcity. Therefore, we propose a novel pruning method to release less prioritized adversarial perturbations with minimal impact on the absolute model output variances, thereby creating space to introduce perturbations that facilitate dodging attacks.

Our proposed Adv-Pruning and be combined with various adversarial attacks. In the following, we will introduce our proposed Adv-Pruning based on Lagrangian attack in detail. In the Priming Stage, we utilize Eq. (3) as the Priming loss $\mathcal{L}^p$ to craft the adversarial face examples:

$$x_t^{adv} = \prod_{x^s,\epsilon}\left(x_{t-1}^{adv} - \beta\text{sign}\left(\nabla_{x_{t-1}^{adv}}\mathcal{L}^p\right)\right) \tag{4}$$

where $t$ is the iteration of the optimization process of adversarial examples, and $\beta$ is the step size when optimizing the adversarial face examples in the Priming stage, and $\prod(x)$ is the projection function that projects $x$ onto the $L_p$ norm bound.

**Adversarial Priority Quantification.** After completing the Priming Stage, we obtain an adversarial example with varying magnitudes of gradient perturbations across different regions. Following this, we proceed to the Pruning stage to process the crafted adversarial example. In order to prune the adversarial perturbation, our initial step is to assess its priority. To estimate this priority, we propose Adversarial Priority Quantification to quantify the priority of adversarial perturbations. Specifically, Adversarial Priority Quantification utilizes the magnitude of the adversarial perturbation as a measure. A lower magnitude implies a lesser impact on the performance of the adversarial examples generated after sparsification, as the supremum of absolute model output variances is directly proportional to the magnitude of the adversarial perturbations. The proof is *in the supplementary*.

Let the adversarial examples be $x^{adv}$. The formula to calculate the priority can be expressed as:

$$\mathcal{I} = |x^{adv} - x^s| \tag{5}$$

where $\mathcal{I} \in \mathbb{R}^{\text{CHW}}$. C, H, and W are the channel number, height, and width of the face images, respectively.

Once the priority values of the adversarial perturbations are quantified, we employ these values to release less prioritized adversarial perturbations. Let $\kappa$ be the sparsity ratio for pruning the adversarial face examples that measure the ratio of perturbations to be set into zero. Let $s = $ CHW be the number of adversarial perturbation elements. We arrange the elements in a flattened vector of $\mathcal{I}$ in ascending order (from the lowest to the highest):

$$Q = \text{Sort}\left(\Psi\left(\mathcal{I}\right)\right) \tag{6}$$

where $\Psi$ is the flatten operation.

Let $\mathcal{W}$ be the set of the elements of the adversarial perturbations to be pruned. Given the priority calculation method for pruning,

the value of $\mathcal{W}$ can be calculated as follows:

$$\mathcal{W} = Q\left[: \kappa s\right] \tag{7}$$

where the colon denotes the slice operation to obtain the first $\kappa s$ elements. The pruning mask, which has the same shape as $\mathcal{I}$, can be obtained by utilizing $\mathcal{W}$:

$$\mathcal{M}_{i,j,k} = \begin{cases} 0, & \text{if } \mathcal{I}_{i,j,k} \in \mathcal{W} \\ 1, & \text{if } \mathcal{I}_{i,j,k} \notin \mathcal{W} \end{cases} \tag{8}$$

By utilizing the mask, we can apply the following formula to prune the adversarial example:

$$\bar{x}^{adv} = x^s + \left(x^{adv} - x^s\right) \odot \mathcal{M} \tag{9}$$

where $\bar{x}^{adv}$ is the adversarial face example after pruning.

**Biased Gradient Adaptation.** During the Restoration stage, we restore the adversarial face examples in the previously pruned region using our proposed Biased Gradient Adaptation. Biased Gradient Adaptation using the following loss function to craft gradient biased to the dodging attacks to adapt the crafted adversarial examples into the space that favors dodging attacks.

$$\mathcal{L}^r = \tilde{\lambda}\mathcal{L}^i + \mathcal{L}^d \tag{10}$$

where $\tilde{\lambda}$ is a weight that is lower than $\lambda$ that is objective for crafting adversarial face examples that favor dodging attacks. The mask representing the regions for restoring the adversarial examples can be denoted as:

$$\mathcal{A} = 1 - \mathcal{M} \tag{11}$$

Subsequently, we utilize the following formula to restore the pruned adversarial face examples:

$$x_t^{adv} = \prod_{x^s, \epsilon} \left(x_n^{adv} + \mathcal{A} \odot \left(x_{t-1}^{adv} - \gamma \text{sign}\left(\nabla_{x_{t-1}^{adv}}\mathcal{L}^r\right) - \bar{x}^{adv}\right)\right) \tag{12}$$

where $\gamma$ is the step size when optimizing the adversarial face examples in the Restoration stage, $\bar{x}^{adv}$ is the adversarial example crafted by the Priming Stage, and $\nabla_{x_{t-1}^{adv}}\mathcal{L}^r$ is the biased gradient. The pseudo-code of our proposed method based on the Lagrangian attack is illustrated *in the supplementary.*

# 4 EXPERIMENTS

## 4.1 Experimental Setting

**Datasets.** Face images play a pivotal role in multimedia processing applications. Therefore, the research on adversarial attacks on FR has a significant impact on security and privacy in multimedia processing. We opt to use the LFW [19], CelebA-HQ [24], and FFHQ [25] datasets for our experiments. LFW serves as an unconstrained face dataset for FR. CelebA-HQ and FFHQ consists of high-quality images. The LFW and CelebA-HQ utilized in our experiments are identical to those employed in [70, 71], while FFHQ is the corresponding dataset provided by the Sibling-Attack official page, ensuring the consistency for analysis.

**Face Recognition Models.** The normal trained FR models employed in our experiments include IR152 [16], FaceNet [41], Mobile-Face (abbreviated as MF) [9], ArcFace [9], CircleLoss [45], CurricularFace [21], MagFace [36], MV-Softmax [52], and NPCFace [66]. IR152, FaceNet, and MF are identical to those used in [18, 63, 70, 71]. ArcFace, CircleLoss, CurricularFace, MagFace, MV-Softmax, and NPCFace are the official models available in FaceX-ZOO [23]. Additionally, we incorporate adversarial robust FR models in our experiments, denoted as IR152$^{adv}$, FaceNet$^{adv}$, and MF$^{adv}$, which are identical to those used in [71]. For calculating the ASR in impersonation and dodging attacks, we choose the thresholds based on FAR@0.001 on the entire LFW dataset.

**Attack Setting.** Without any particular emphasis, we set the maximum allowable perturbation magnitude to 10 based on the $L_\infty$ norm bound and utilize the Lagrangian attack method as the attack in both the Priming and Restoration stages. Additionally, we specify the maximum number of iterative steps as 200. For both the Priming and Restoration stages, the step size is uniformly designated as 1.0.

**Evaluation Metrics.** We employ Attack Success Rate (ASR) to evaluate the performance of various attacks. ASR signifies the proportion of successfully attacked adversarial examples out of all the adversarial examples. We use ASR$^i$ and ASR$^d$ to denote impersonation and dodging ASR, respectively. The detailed calculation methods for ASR$^i$ and ASR$^d$ are provided *in the supplementary.*

**Compared methods.** Our proposed attack is a restricted attack method that aims to maliciously attack FR systems to expose more blind spots of them. It is not fair to compare our proposed method

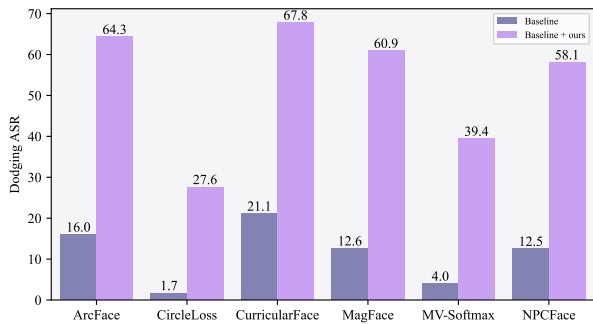

**Figure 3: The ASR on FR models trained by multiple algorithms.**

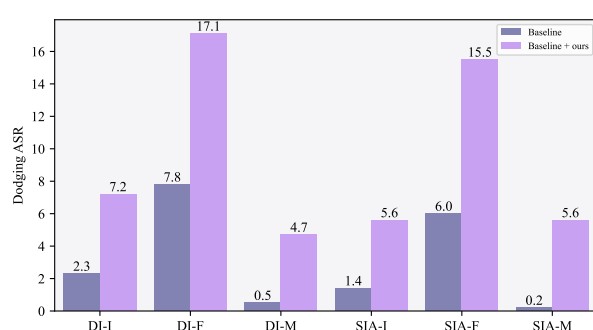

**Figure 4: Comparisons of ASR (%) on LFW with adversarial robust models as victim models.**

**Table 1: Comparisons of dodging ASR (%) results for attacks on the LFW and CelebA-HQ datasets. The surrogate models are presented in the first column, and the victim models are listed in the second row. The numbers before and after the slash represent the results of the baseline attack and the attack that combines the baseline attack with our proposed attack, respectively.**

| Surrogate Model | Attack | LFW | | | CelebA-HQ | | |
|---|---|---|---|---|---|---|---|
| | | IR152 | FaceNet | MF | IR152 | FaceNet | MF |
| IR152 [16] | DI | 95.4 / **100.0** | 5.8 / **11.3** | 0.2 / **0.8** | 87.9 / **100.0** | 8.4 / **16.4** | 0.4 / **1.2** |
| | VMI | 92.7 / **100.0** | 17.3 / **32.5** | 1.2 / **9.5** | 92.2 / **100.0** | 14.3 / **27.9** | 1.2 / **4.1** |
| | SSA | 78.8 / **100.0** | 5.7 / **22.5** | 0.9 / **10.6** | 83.8 / **99.9** | 7.2 / **19.6** | 0.4 / **5.6** |
| | DFANet | 98.9 / **100.0** | 1.4 / **4.2** | 0.0 / **0.3** | 98.9 / **100.0** | 2.3 / **6.0** | 0.0 / **0.4** |
| | SIA | 81.7 / **100.0** | 13.0 / **37.5** | 0.8 / **8.9** | 78.4 / **100.0** | 13.2 / **35.6** | 0.7 / **7.4** |
| | BSR | 52.4 / **100.0** | 5.3 / **17.6** | 0.1 / **1.5** | 48.5 / **99.9** | 5.4 / **18.0** | 0.3 / **1.9** |
| | BPFA | 92.6 / **100.0** | 1.7 / **7.3** | 0.0 / **1.2** | 90.4 / **100.0** | 2.1 / **8.1** | 0.1 / **0.8** |
| FaceNet [41] | DI | 5.3 / **10.3** | 99.8 / **99.9** | 3.1 / **10.3** | 1.5 / **3.1** | 99.4 / **99.9** | 1.8 / **4.7** |
| | VMI | 9.7 / **14.3** | 99.8 / **99.9** | 6.2 / **13.2** | 3.1 / **7.1** | 99.3 / **99.8** | 3.6 / **9.3** |
| | SSA | 6.0 / **14.0** | 97.5 / **99.9** | 6.6 / **26.2** | 2.0 / **5.5** | 96.9 / **99.7** | 4.2 / **14.6** |
| | DFANet | 1.6 / **3.3** | 99.8 / **99.9** | 0.4 / **2.7** | 0.5 / **2.6** | 99.1 / **100.0** | 0.8 / **4.1** |
| | SIA | 11.2 / **20.6** | 99.5 / **99.9** | 8.7 / **21.2** | 4.0 / **8.9** | 99.4 / **99.9** | 5.4 / **13.7** |
| | BSR | 12.2 / **19.2** | 98.6 / **99.9** | 9.0 / **17.8** | 4.6 / **10.1** | 98.8 / **99.9** | 5.3 / **14.1** |
| | BPFA | 4.7 / **16.8** | 98.6 / **100.0** | 1.6 / **15.0** | 1.1 / **4.2** | 99.0 / **100.0** | 0.6 / **5.1** |
| MF [9] | DI | 2.2 / **7.3** | 18.2 / **36.4** | 99.2 / **100.0** | 0.1 / **2.5** | 12.1 / **31.3** | 95.2 / **100.0** |
| | VMI | 1.0 / **2.8** | 8.4 / **20.9** | 99.7 / **100.0** | 0.2 / **0.4** | 5.2 / **15.0** | 98.2 / **100.0** |
| | SSA | 0.7 / **4.1** | 6.1 / **23.5** | 98.3 / **100.0** | 0.0 / **0.6** | 3.9 / **18.5** | 93.3 / **100.0** |
| | DFANet | 0.2 / **1.0** | 1.5 / **5.8** | 99.6 / **100.0** | 0.0 / **0.2** | 1.1 / **7.7** | 99.1 / **100.0** |
| | SIA | 1.0 / **5.9** | 10.6 / **36.6** | 98.4 / **100.0** | 0.1 / **2.4** | 9.0 / **24.4** | 96.3 / **100.0** |
| | BSR | 0.4 / **1.5** | 3.7 / **14.7** | 84.9 / **100.0** | 0.1 / **0.6** | 2.9 / **12.6** | 77.6 / **100.0** |
| | BPFA | 0.9 / **4.1** | 4.6 / **20.4** | 97.7 / **100.0** | 0.0 / **2.3** | 4.0 / **20.4** | 96.2 / **100.0** |

**Table 2: Comparisons of ASR (%) with multi-task attacks on LFW dataset. Models in the second row are victim models.**

| Attack | $ASR^d$ | | | $ASR^i$ |
|---|---|---|---|---|
| | IR152 | FaceNet | MF | |
| Lagrangian | 3.9 | 26.5 | **100.0** | 26.0 |
| Lagrangian + ours | **7.3** | **36.4** | **100.0** | **26.6** |
| DA | 11.0 | 35.6 | 99.1 | 37.4 |
| DA + ours | **17.5** | **44.9** | **99.4** | **37.8** |

with unrestricted attacks that do not limit the magnitude of the adversarial perturbations. Therefore, we choose restricted attacks on FR that aim to maliciously attack FR systems [69] [71] [29] and state-of-the-art transfer attacks [59] [33] [53] [50] as our baseline.

## 4.2 Comparison Study

We compare our proposed attack method with the state-of-the-art attacks on multiple FR models and datasets. Several adversarial examples are illustrated in Fig. 5. The attack performance results are shown in Table 1. Table 1 illustrates that the incorporation of our proposed attack method significantly enhances the dodging ASR of adversarial attacks. It is worth noting that the average black-box impersonation ASRs of the baseline attacks in Table 1 also increase after integrating our proposed attack method. This demonstrates the effectiveness of our proposed method in improving the dodging attack performance while simultaneously maintaining the

impersonation attack performance. Furthermore, we conducted a comparison between our proposed Adv-Pruning and multi-task attacks using MF as the surrogate model on LFW based on DI. For our proposed method, we choose the corresponding multi-task attack as the attack for both the Priming and Restoration stages. The dodging ASR and average black-box impersonation ASR results are shown in Table 2. Table 2 underscores the effectiveness of our method in enhancing the dodging performance of multi-task attacks while maintaining the impersonation performance. To further validate our proposed attack method on additional FR models, we selected SIA [53] as Baseline and IR152 as the surrogate model. The experimental settings are consistent with those described in Table 1. The dodging ASR across multiple FR models is demonstrated in Fig. 3. As depicted in Fig. 3, the dodging ASR improves on multiple FR models after integrating our proposed method, further confirming the effectiveness of our attack.

In practical application scenarios, victims can employ adversarial robust models to defend against adversarial attacks. Consequently, it becomes crucial to evaluate the performance of adversarial attacks on these robust models. In this study, we generate adversarial examples on the LFW dataset using MF as the surrogate model and assess the performance of various attacks on the adversarial robust models. The results are presented in Fig. 4. The letters following the en dash represent the surrogate models, with 'I', 'F', and 'M' corresponding to $IR152^{adv}$, $FaceNet^{adv}$, and $MF^{adv}$, respectively. Fig. 4 illustrates that the inclusion of our proposed method leads to improvements in both dodging and impersonation performance.

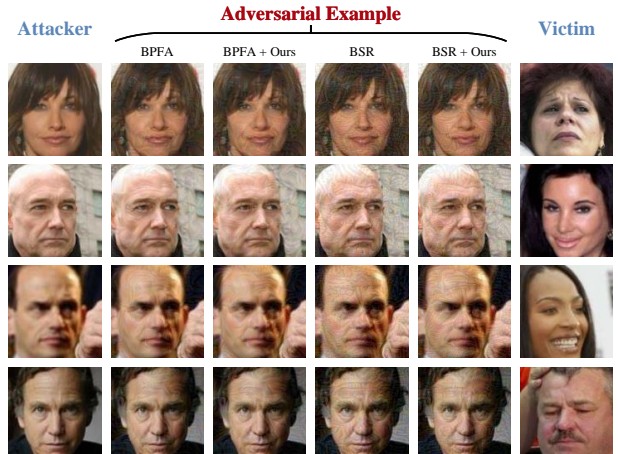

**Figure 5: The Illustration of adversarial examples crafted by various attacks. First column: some attacker images. Last column: the corresponding victim images. The second to fifth columns exhibit the corresponding adversarial face examples crafted by BPFA, BPFA + ours, BSR, and BSR + ours, respectively.**

These results serve as evidence of the effectiveness of our proposed method on adversarial robust models.

JPEG compression is a widely adopted method for image compression during transmission, concurrently acting as a defense mechanism against adversarial examples. To assess the effectiveness of our proposed attack under JPEG compression, we utilize DI as the baseline attack and MF as the surrogate model, evaluating the attack performance on ArcFace and CurricularFace models with experimental settings consistent with those described in Table 1. The results are illustrated in Fig. 6. These results demonstrate that across varying levels of JPEG compression, our proposed attack method consistently outperforms the baseline attack, thereby highlighting its effectiveness under JPEG compression.

The experimental results on negative cosine similarity loss, and Sibling-Attack are presented *in the supplementary*.

### 4.3 Ablation Study

To delve into the properties of our proposed attack method, we conducted an ablation experiment using DI as the Baseline attack, with MF serving as the surrogate model on the LFW dataset. To confirm the effectiveness of our pruning method, we employed the Random Zeroing (RZ) method, which randomly sets adversarial perturbations to zero. We applied this method and our pruning method to free up 20% of the adversarial perturbations crafted by the Lagrangian attack. For *Fine-tuning*, we employed the Lagrangian attack as the method to further optimize the Lagrangian adversarial examples with a lower $\lambda$. The dodging attack ASR and average black-box impersonation ASR results are shown in Table 3. Table 3 demonstrates that our proposed pruning method for adversarial examples achieves a significantly smaller decrease in ASR than *RZ* after pruning 20% of adversarial perturbations, indicating the effectiveness of our pruning method. After being processed using the Pruning and Restoration stage of our proposed Adv-Pruning

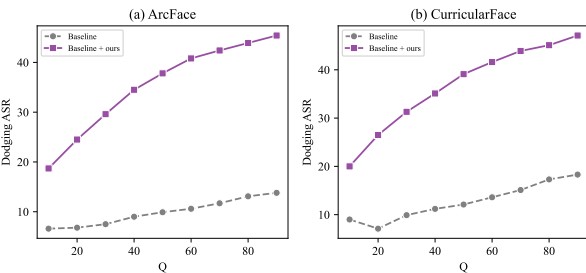

**Figure 6: The dodging ASR under various JPEG Q values.**

method, both impersonation and dodging ASR of the crafted adversarial face examples are recovered and higher than the Baseline attack method. These results demonstrate the effectiveness of our proposed Adv-Pruning in improving the dodging performance of adversarial attacks on FR without compromising the impersonation attack performance.

The sparsity ratio quantifies the proportion of adversarial perturbations that are allowed to be discarded during the Pruning stage. This ratio greatly impacts the performance of our proposed attack method. Hence, we conducted a sensitivity study on the sparsity ratio to analyze its effect on performance of the algorithm. The Lagrangian attack method based on DI is selected as the Baseline. We conduct a hyperparameter sensitivity study on LFW using FaceNet as the surrogate model, and adjust the value of $\tilde{\lambda}$ to ensure that the average black-box impersonation ASR results were within a 0.4% absolute difference compared to the Baseline. The dodging ASR results are illustrated in the right plot of Fig. 7. The results illustrate that the dodging ASR of our proposed method initially increases and then decreases as the sparsity ratio increases. When the sparsity ratio increases, a greater number of adversarial perturbations are pruned, creating more empty regions for the adversarial perturbations that favor dodging attacks in the Restoration stage. If the sparsity ratio is set to a too-high value, an excessive number of adversarial perturbations are allowed to be freed up, resulting in a degradation of performance for the adversarial examples crafted by the Priming stage. Consequently, the performance of adversarial face examples will decrease.

### 4.4 Analytical Study

**Multi-identity Samples among the Natural Face Images:** Multi-identity samples are intriguing samples that can be classified as multiple classes in FR. In this section, we will explore the existence of multi-identity samples among the natural face images. We randomly select negative face pairs from the entire LFW dataset. Subsequently, we use MF as our FR model to extract the embeddings of the face images in each face pair and calculate the cosine similarity between the two images. If the cosine similarity surpasses the threshold, both images in the pair are classified as belonging to multiple identities, indicating that they are multi-identity samples. Our findings demonstrate the presence of multi-identity samples among the benign face images, as illustrated in the bottom left of Fig. 1. The multi-identity samples in Fig. 1 closely resemble the appearances of the identities they are classified into.

**Table 3: Comparisons of ASR (%) results of dodging attack and impersonation attacks on the LFW dataset. The models in the second row are the victim models.**

| | ASR$^d$ | | | ASR$^i$ |
|---|---|---|---|---|
| Attack | IR152 | FaceNet | MF | |
| Baseline | 2.2 | 18.2 | 99.2 | 25.6 |
| Lagrangian | 3.9 | 26.5 | **100.0** | 26.0 |
| *Fine-tuning* | 4.2 | 26.3 | **100.0** | 25.6 |
| *RZ* | 0.6 | 4.4 | 94.8 | 15.1 |
| *Pruning* | 3.4 | 24.8 | **100.0** | 25.4 |
| Adv-Pruning | **5.4** | **32.5** | **100.0** | **26.3** |

To analyze the cause of this phenomenon, we need to consider the properties of both the multi-identity samples and the FR model. Commonly-used FR models are well-trained and capable of correctly classifying the majority of benign face images. However, there are some benign face images that the FR model fails to classify accurately, and these samples are referred to as hard samples [20, 71]. Multi-identity samples are a specific type of hard sample known as hard negative samples. Typically, hard negative samples exhibit a similar appearance [66], indicating that the multi-identity samples among the benign face images share a resemblance.

**Universality of Multi-identity Samples among Adversarial Face Examples**: The previous impersonation methods craft adversarial face examples only use the impersonation loss $\mathcal{L}^i$. In this section, we will investigate the ratio of multi-identity samples and evaluate the effectiveness of previous impersonation methods in terms of dodging attacks. We utilize the Multi-identity Sample Ratio (MSR) to gauge the proportion of multi-identity samples in the adversarial face examples capable of executing successful impersonation attacks. These multi-identity samples can be recognized as both the attacker and victim identities in the setting of our paper. The detailed calculation method of MSR is *in the supplementary*. We evaluate the MSR, impersonation ASR and dodging ASR using MF as the surrogate model on the LFW dataset and the results are demonstrated *in the supplementary*. The results demonstrates that most of the crafted adversarial face examples are multi-identity samples in the black-box setting. This indicates that most adversarial face examples generated through previous impersonation attacks are unable to attain a successful dodging attack in the black-box setting. Nevertheless, in the white-box setting, the majority of adversarial face examples capable of executing successful impersonation attacks also demonstrate success in dodging attacks.

To analyze the reason, we consider the metric used to determine whether two face images belong to the same identity in FR. Since FR is an open-set task, we rely on the distance in the embedding space to make decisions. The top of Fig. 1 illustrates two decision boundaries for each FR model, one for attacker identity and one for victim identity. In the white-box setting, if we generate adversarial examples using $\mathcal{L}^i$, these adversarial examples can penetrate a space where they are recognized as the victim identity rather than the attacker identity. However, the decision boundary of the black-box model differs from that of the surrogate model. In the black-box setting, most adversarial examples are found between

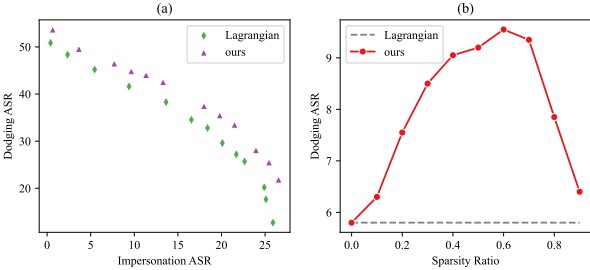

**Figure 7: (a) The trade-off between the ASR (%) of impersonation attack and dodging attack of adversarial examples. (b) The dodging ASR (%) in different sparsity ratios.**

the decision boundaries of the black-box model, resulting in the majority of adversarial examples crafted using $\mathcal{L}^i$ being multi-identity samples. This demonstrates that the adversarial face examples are positioned near the decision boundary in the black-box setting.

**The Trade-off Between the Impersonation Attacks and Dodging Attacks**: Owing to the inherent conflict during the optimization process of impersonation and dodging losses in the black-box setting, there exists a trade-off between impersonation and dodging performance. We craft adversarial face examples using Lagrangian attack and our proposed Adv-Pruning on LFW based on DI. The average black-box results are demonstrated in the left plot of Fig. 7.

The results illustrate that our proposed method can reduce the trade-off between impersonation and dodging performance in the black-box setting. The pruning operation of our proposed Adv-Pruning serves to sparsify the adversarial perturbations while preserving the impersonation performance. On the other hand, the restoration operation tends to introduce adversarial perturbations in the pruned areas, specifically favoring dodging attacks. These operations effectively enhance the dodging attack performance while maintaining the impersonation attack performance, ultimately mitigating the trade-off.

## 5 CONCLUSION

In this paper, we delve into the issue of multi-identity samples among adversarial face examples. Our research reveals the universality of multi-identity samples among adversarial face examples crafted by previous impersonation attacks and the success of an impersonation attack may not necessarily imply the success of dodging attacks on FR systems in the black-box setting. In order to improve dodging performance without compromising impersonation performance, we proposed a novel attack, namely Adv-Pruning. Adv-Pruning comprises Priming, Pruning, and Restoration Stages. Leveraging our proposed Adversarial Priority Quantification, we identify less prioritized adversarial perturbations with minimal impact on absolute model output variances. Through our proposed Biased Gradient Adaptation, biased gradient perturbations are applied to the sparsified regions, adapting adversarial face examples to a space favoring evasion attacks. Extensive experiments demonstrate the effectiveness of our proposed method.

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
