# OpenReview forum: "Rethinking Impersonation and Dodging Attacks on Face Recognition Systems"
_acmmm.org/ACMMM/2024/Conference — MM2024 Poster_

### Official Review · Reviewer_a7Fm · 2024-05-23

**Rating:** 3
**Confidence:** 3

**Summary:**

The authors propose adversarial pruning strategies to achieve guaranteed target attack while improving the performance of untargeted attacks.

**Strengths:**

1. The idea is Interesting.
2. Overall, the technique is feasible.

**Limitations:**

1. Writers write their papers with a lot of irrelevant adjectives that do not enhance the scholarship of the paper but tend to create ambiguity. For example, benign, blind spots, endeavors, open-set.
2. The biggest role of the diagram placed in the introduction is to allow the reader to quickly capture the author's motivation, which requires a low level of complexity in the content of that diagram. And the corresponding textual explanation should not bring up new questions and should be a general description of the figure. However, the author's Figure 1 and the second paragraph do not form an effective contrast. The text of the second paragraph does not reasonably describe the content of Figure 1, but further introduces new questions. The interpretation of "multiple identities" is ambiguous.
3. As far as I know, a corresponding targetless attack task (evasion attack) has already been proposed in the black box task, and the conclusions drawn by the authors in line 149 are not rigorous enough nor backed up by corresponding references. Please refer the author to the following paper.
Guo C, Gardner J, You Y, et al. Simple black-box adversarial attacks[C]//International conference on machine learning. PMLR, 2019: 2484-2493.
Andriushchenko M, Croce F, Flammarion N, et al. Square attack: a query-efficient black-box adversarial attack via random search[C]//European conference on computer vision. Cham: Springer International Publishing, 2020: 484-501.
Zhou L, Cui P, Zhang X, et al. Adversarial eigen attack on black-box models[C]//Proceedings of the IEEE/CVF conference on computer vision and pattern recognition. 2022: 15254-15262.
4. The background of this paper emphasizes face research and states at the innovation point that it is the first research in face perspective, but the analysis of the proposed scheme lacks a close integration with face and is more like traditional adversarial sample research. Because traditional adversarial samples have a lot of research on targetless attacks from a black box perspective, this makes the author's innovation unconvincing.
5. The title of this paper is “rethinking”, and I would have preferred to see the author's mathematical model or defense model for the generation of adversarial samples, but the fact that the author is still essentially generating adversarial samples and that the author doesn't analyze in depth the mathematics between targeted and untargeted attackers makes the title seem large and vague in comparison to the content.
6. I am curious why why the authors did not show the visual evaluation metrics such as PSNR, SSIM, FID, LPIPS, etc. before and after the success of the counterattack in the experimental section. Because it would be more convincing to show visual quality metrics even if infinite paradigms are used. Please refer to the following paper.
Chen Z, Wang Z, Huang J J, et al. Imperceptible adversarial attack via invertible neural networks[C]//Proceedings of the AAAI Conference on Artificial Intelligence. 2023, 37(1): 414-424.
7. I would suggest that the author's supplementary material does not need to be resubmitted in the body of the text, just the appropriate supplementary material
8. This paper in the introduction and the innovation point of the affirmation mainly lies in the simultaneous realization of targetless and targeted attacks, but the author's experimental setup is more like a reinforcement module, specifically this paper ultimately want to optimize the performance of the existing algorithms rather than breakthroughs to complete the targeted and untargeted attacks, for the simultaneous realization of targeted and untargeted attacks the amount of experiments to be far less than the reinforcement module, which makes the whole paper's fragmentation This makes the whole thesis more split.

**Suitability:**

3

---

### Official Review · Reviewer_KXmz · 2024-05-24

**Rating:** 3
**Confidence:** 2

**Summary:**

This paper presents Adversarial Pruning designed to enhance dodging capabilities of adversarial face examples while preserving their impersonation capabilities through a three-stage process of priming, pruning, and restoration.

**Strengths:**

1. The paper identifies the multi-identity problem arising from adversarial examples optimized solely for impersonation attacks on face recognition systems.

2. The method employs a three-stage process (priming, pruning, and restoration) to comprehensively optimize the decision boundary perturbation.

3. Experiment results seem to be promising in terms of both dodging and impersonation success rates.

**Limitations:**

1.The Adv-Pruning method consists of three stages (priming, pruning, and restoration), making the process relatively complex and potentially increasing implementation difficulty. Is there an opportunity to simplify these components? Additionally, the authors may consider releasing the source code upon paper acceptance. While the intricate process of adversarial example movement across decision boundaries is interesting, providing further theoretical or empirical evidence to intuitively describe this process beyond the attack's effectiveness would offer more insight.

2.Although the proposed method improves attack success rates on adversarially robust models, defenders have various strategies to counter adversarial attacks. For example, the perturbations produced with Adv-Pruning appear somewhat more noticeable, according to the illustrations. Does this imply that the perturbations are more detectable? Moreover, are these perturbations more vulnerable to purification or input transformation defenses, such as advanced adversarial purification based on diffusion models?

3.Given the complexity of the optimization process, the resulting perturbations might be subtle. Is it feasible to implement the proposed attacks in the physical domain rather than the digital domain? For instance, can this method be applied to create patch-based perturbations or confined to specific regions to produce adversarial glasses or other accessories to implement practical attacks in the real world?

**Suitability:**

2

---

### Official Review · Reviewer_Ukwt · 2024-05-25

**Rating:** 3
**Confidence:** 3

**Summary:**

This paper explores the issues of impersonation and dodging attacks in facial recognition systems (FR). Existing methods typically focus on impersonation attacks while overlooking the potential failure of dodging attacks when impersonation attacks succeed in a black-box setting. To address both attacks simultaneously, the authors propose a novel method called Adversarial Pruning (Adv-Pruning). This method enhances the dodging capability of adversarial samples while maintaining their impersonation ability through three stages: Priming, Pruning, and Restoration.

**Strengths:**

See Weakness

**Limitations:**

1. The paper lacks ethical considerations. The authors do not propose a feasible defense mechanism or applicable scenarios. Is the purpose of proposing attack methods on facial recognition systems solely for malicious intent?

2. If impersonation is merely a targeted attack and dodging is an untargeted attack, this work lacks novelty. Both impersonation (targeted attacks) and dodging (untargeted attacks) have been extensively discussed in adversarial attack research. Simply combining these two attacks may not constitute a significant research contribution.

3. The authors claim to achieve both impersonation and dodging attacks, but the descriptions of these two attributes are unclear.

4. According to the paper, in a black-box setting, achieving a successful impersonation attack might negatively affect the effectiveness of the dodging attack. Are these two attributes negatively correlated?

5. The attacks proposed by the authors are not sufficiently stealthy. The perturbation magnitude is set to 10, which is quite high for adversarial attacks. Observing these results with the naked eye reveals noticeable changes. The authors should quantify the impact of adversarial perturbations on image quality using metrics such as PSNR. This would provide a more objective assessment of the perturbation’s visibility.

6. The experimental results indicate that the Adv-Pruning method outperforms existing adversarial attack methods across various models, but the specific experimental settings and result analysis might be inadequate.

**Suitability:**

2

---

### Official Review · Reviewer_dt2U · 2024-05-27

**Rating:** 4
**Confidence:** 4

**Summary:**

The authors found that adversarial examples capable of impersonation attacks do not necessarily possess the ability to perform dodging attacks. Consequently, the authors posed the question of how to generate adversarial examples that simultaneously have both dodging and impersonation attack capabilities. To address this, the authors proposed a three-stage attack method called Adv-pruning, which consists of priming, pruning, and Restoration. Experimental results demonstrate that this method significantly outperforms the baseline.

**Strengths:**

1. The problem of generating adversarial examples that possess both impersonation and dodging attack capabilities is valuable. However, I believe that the scenarios considered by the authors require more than just this (see weakness).

2. Extensive experiments have demonstrated the superiority of Adv-Pruning over existing methods.

**Limitations:**

1. Some hyperparameter values are not provided, such as $\lambda$ and $\tilde{\lambda}$. In particular, $\tilde{\lambda}$, being a newly introduced parameter in this paper, requires corresponding ablation experiments. This parameter also makes some comparative experiments confusing. For example, how was $\lambda$ chosen for the fine-tuning in Table 3? Was it consistent with the proposed method, or was an optimal value achieved?

2. The preprocessing of pruning is an important contribution of this paper, yet in Table 3, the authors only compared with RZ (random zeroing) and concluded that pruning leads to less performance degradation than RZ, which is not surprising. Our expectation for the pruning step is not to reduce the drop in attack success rate, but to hope that fine-tuning after pruning can achieve a higher attack success rate. Therefore, the authors should not just compare RZ with pruning, but rather fine-tune after RZ and compare with the proposed method, selecting an optimal RZ ratio. Additionally, the pruning method proposed by the authors is heuristic; are there better pruning methods? For instance, using gradient information of the loss as the importance of each pixel might be better than noise magnitude. Furthermore, [1] proposed to mask noise in the facial key point regions at each iteration, which is similar to pruning based on facial key points and is more effective than no pruning or random pruning. Can this approach be borrowed?

3. Visualization of the pruning regions would be helpful. I am curious about whether the final pruning regions are primarily important areas (such as facial key points) or unimportant areas (background).

4. I am curious about the application scenarios of the problem studied in this paper. Dodging attacks and impersonation attacks each have their own application scenarios. Impersonation attacks can be used to infiltrate someone else's facial recognition system on a phone, where the effect of dodging is not important. Dodging attacks can be used to protect one's privacy, without needing to consider the effect of impersonation. The authors mentioned in the Introduction: "individuals with malicious intent are prone to creating adversarial face examples incorporating their own facial features to manipulate FR systems to mistakenly identify them as pre-defined victims during impersonation attacks. Concurrently, the individuals strive to evade accurate identification as perpetrators, thereby circumventing detection and preventing legal accountability." I acknowledge that in such scenarios, both dodging and impersonation need to be achieved, but it goes beyond that. There should be an additional requirement in this scenario: The FR system features an internal facial database, where adversarial examples must not only surpass a similarity threshold with the target individual but also demonstrate a higher similarity than that with any other faces in the database. Otherwise, the authors' claimed goal cannot be achieved. A similar setting was considered in [62] (see [62]'s rank-N-T and rank-N-UT). I suggest the authors also report such metrics, although this might be a much more challenging problem.

[1] Yang, X., Yang, D., Dong, Y., Su, H., Yu, W., & Zhu, J. (2020). Robfr: Benchmarking adversarial robustness on face recognition. arXiv preprint arXiv:2007.04118.

**Suitability:**

2

---

### Meta-Review · Area_Chair_GMDh · 2024-06-29

**Recommendation:** Accept (Poster)
**Confidence:** 4

**Metareview:**

The authors discovered that adversarial examples effective for impersonation attacks might not be capable of dodging attacks. They introduced a three-stage attack method called Adv-pruning, which includes priming, pruning, and restoration, to generate examples that can perform both attacks. Experimental results show that this method significantly outperforms the baseline. The authors are better to further clarify the definitions about Impersonation Attack  and Dodging Attack.